# Phase diagrams guide synthesis of highly ordered intermetallic electrocatalysts: separating alloying and ordering stages

Wei-Jie Zeng[1], Chang Wang[2], Qiang-Qiang Yan [1], Peng Yin[1], Lei Tong[1] & Hai-Wei Liang [1]✉

Supported platinum intermetallic compound catalysts have attracted considerable attention owing to their remarkable activities and durability for the oxygen reduction reaction in proton-exchange membrane fuel cells. However, the synthesis of highly ordered intermetallic compound catalysts remains a challenge owing to the limited understanding of their formation mechanism under high-temperature conditions. In this study, we perform in-situ high-temperature X-ray diffraction studies to investigate the structural evolution in the impregnation synthesis of carbon-supported intermetallic catalysts. We identify the phase-transition-temperature ($T_{PT}$)-dependent evolution process that involve concurrent (for alloys with high $T_{PT}$) or separate (for alloys with low $T_{PT}$) alloying/ordering stages. Accordingly, we realize the synthesis of highly ordered intermetallic catalysts by adopting a separate annealing protocol with a high-temperature alloying stage and a low-temperature ordering stage, which display a high mass activity of 0.96 A $mg_{Pt}^{-1}$ at 0.9 V in $H_2-O_2$ fuel cells and a remarkable durability.

Carbon-supported Pt–M (M is typically a 3d transition metal) ordered intermetallic compounds (IMCs) have been recognized as some of the most promising electrocatalysts for the sluggish oxygen reduction reaction (ORR)[1–6], which is the bottleneck restricting the large-scale commercialization of proton-exchange-membrane fuel cell (PEMFC) vehicles[7–9]. The Pt–M IMCs have defined crystal structures and stoichiometries, in which the occupancy of a specific site by Pt or M atoms is determined[10–12], leading to a pronounced ligand and/or strain effects for enhancing ORR activity[13–18]. Moreover, the ordered structures in Pt–M IMCs also probably improve the stability of catalysts under the harsh test conditions in fuel cell[5,19]. Many scientific efforts have been devoted to the synthesis of Pt–M IMCs for ORR applications[20–26], but most of them neglected the considerably low ordering degrees of the catalysts, which have been proven to be positively correlated with ORR performance[5,19,27,28], especially for catalysts prepared by the industry-relevant impregnation method[29].

According to the phase diagrams of binary alloys, the formation of ordered IMCs with defined stoichiometries is thermodynamically favorable below the phase-transition temperature ($T_{PT}$), while the deviation from the stoichiometric composition and elevated annealing temperatures close to or above $T_{PT}$ would be thermodynamically unfavorable for the disorder-to-order phase transition, which highlights the key roles of the alloy composition and annealing temperature in the synthesis of highly ordered IMC catalysts (Fig. 1a). In the impregnation synthesis, high-temperature annealing not only accelerates the inter-particle atom diffusion for alloying Pt with M to form disordered structures with a target stoichiometry, but also promotes the intra-particle diffusion for ordering in the kinetics[11]. Taking PtFe, PtCo, and PtNi as examples (Fig. 1b–d), the annealing temperature of 900 °C for alloying Pt with M is thermodynamically favorable for the generation of intermetallic face-centered tetragonal (fct) PtFe with $T_{PT}$ of ~1300 °C but unfavorable for the generation of fct PtCo ($T_{PT}$ of ~830 °C) and PtNi ($T_{PT}$ of ~630 °C). On the other hand, high-

[1]Hefei National Research Center for Physical Sciences at the Microscale, Department of Chemistry, University of Science and Technology of China, Hefei 230026, China. [2]Dalian Institute of Chemical Physics, Chinese Academy of Sciences, Dalian 116023, China. ✉e-mail: hwliang@ustc.edu.cn

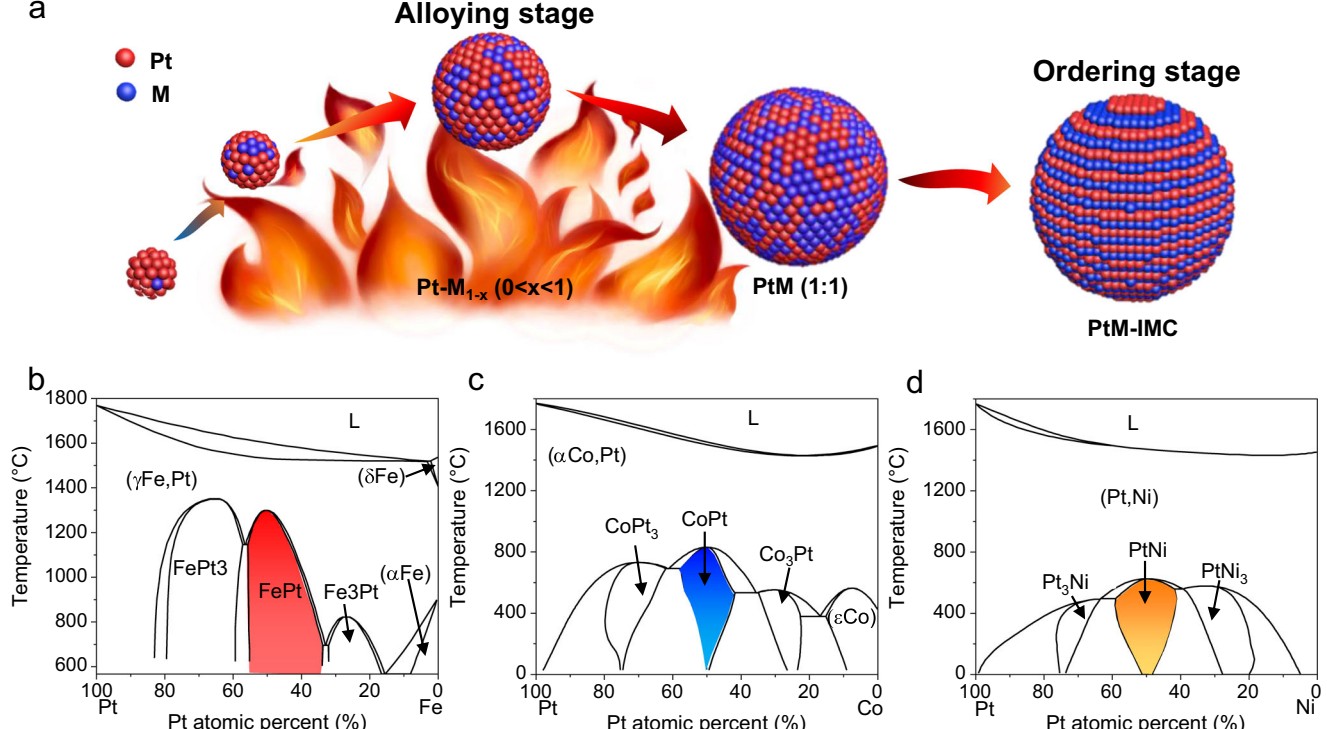

**Fig. 1 | Schematic illustration and binary phase diagrams. a** Schematic illustration of the synthesis of structurally ordered Pt-M (M = Fe, Co, and Ni) intermetallic compounds (IMCs), indicating the separate alloying and ordering stages. **b–d** Binary phase diagrams of Pt–Fe[33], Pt–Co[34], and Pt–Ni[35], respectively. (Reproduced with permission from refs. 33,34, and 35. Copyright 2004, 2019, and 2010 Springer-Verlag.).

temperature annealing treatments often lead to severe thermal sintering and thus undesired larger crystallites with low mass-based activity. Based on these thermodynamic and kinetic fundamentals, the annealing temperature has to be optimized carefully for each IMC synthesis to balance the temperature-dependent trade-off relations between the alloying kinetics, atom ordering kinetics, and metal sintering kinetics.

Herein, we perform in-situ high-temperature X-ray diffraction (HT-XRD) studies to identify the alloying and ordering stages during the synthesis of PtFe, PtCo, and PtNi IMC catalysts by the industrially relevant impregnation method. We find that the alloying and ordering proceed concurrently in the high-temperature heating stage for the PtFe with a high $T_{PT}$, and they occur separately for PtCo and PtNi with low $T_{PT}$. Based on these findings, we realize the synthesis of highly ordered IMC catalysts by separating the high-temperature alloying stage and the low-temperature ordering stage.

## Results

### In situ high-temperature X-ray diffraction (HT-XRD) studies

We first employed in-situ HT-XRD to understand the phase evolution of PtFe, PtCo, and PtNi during the annealing of their corresponding precursors. Carbon black (Black Pearls 2000, abbreviated as BP2000) supported Pt-M precursor powders were obtained by the conventional wet-impregnation method, which were pressed into sheets for the in-situ HT-XRD measurements under the flow of 5 vol% $H_2/N_2$. The HT-XRD data were collected at each temperature during the heating (10 °C min⁻¹), high-temperature holding (850 °C for 2 h), and cooling stages (10 °C min⁻¹). The details of the annealing program are shown in Supplementary Fig. 1a, and the results of HT-XRD are summarized in Fig. 2a–c.

For PtFe, we observed a broad diffraction peak at around 41° when the temperature rose to 300 °C, which corresponded to the (111) plane of the PtFe alloy. This peak became sharp and shifted to a low 2θ

degree gradually as the annealing temperature increased, indicating the growth of the crystal size and alloying process. We further note that a weak diffraction peak at around 33° emerged at 800 °C in the heating stage, which corresponded to the (110) superlattice peak, implying that the formation of fct PtFe occurred. The ordering degree was estimated by comparing the normalized intensity of the (110) peak to the sum of the intensities of the (111) and (200) peaks between the experimental XRD patterns and powder diffraction file (PDF) cards. During the high-temperature holding and cooling stages, more superlattice peaks of the fct PtFe emerged, and their relative intensities increased, corresponding to the gradually increased ordering degree (Fig. 1d). In the case of PtCo (Fig. 2b, e), no superlattice peaks were observed during the heating and high-temperature holding stages; they began to appear until the temperature went down to 800 °C during the cooling stage. For PtNi, we did not observe any superlattice peaks during the whole heating, high-temperature holding, and cooling process (Fig. 2c, f). In other words, we could not obtain ordered fct PtNi structures by conventional annealing synthesis.

From the abovementioned in situ HT-XRD studies, we summarized the different evolution processes of PtFe, PtCo, and PtNi during the same annealing procedure: (i) for PtFe, the alloying and ordering occurred concurrently in the high-temperature heating stage; (ii) for PtCo, the alloying and ordering occurred separately in the heating/holding stages and cooling stage, respectively; and (iii) for PtNi, only alloying occurred during the whole process.

The different evolution processes of PtFe, PtCo, and PtNi could be explained well by their different $T_{PT}$ shown in the phase diagrams (Fig. 1b–d). The $T_{PT}$ of the PtFe (~1300 °C) was much higher than the annealing temperature of the HT-XRD experiments (850 °C). As a result, the disorder-to-order transition was thermodynamically favorable during the whole annealing process. Once the Fe/Pt ratio in some particles approached the stoichiometric values of fct PtFe (i.e., 1:1) during the heating stage, the disordered alloy particles evolved into

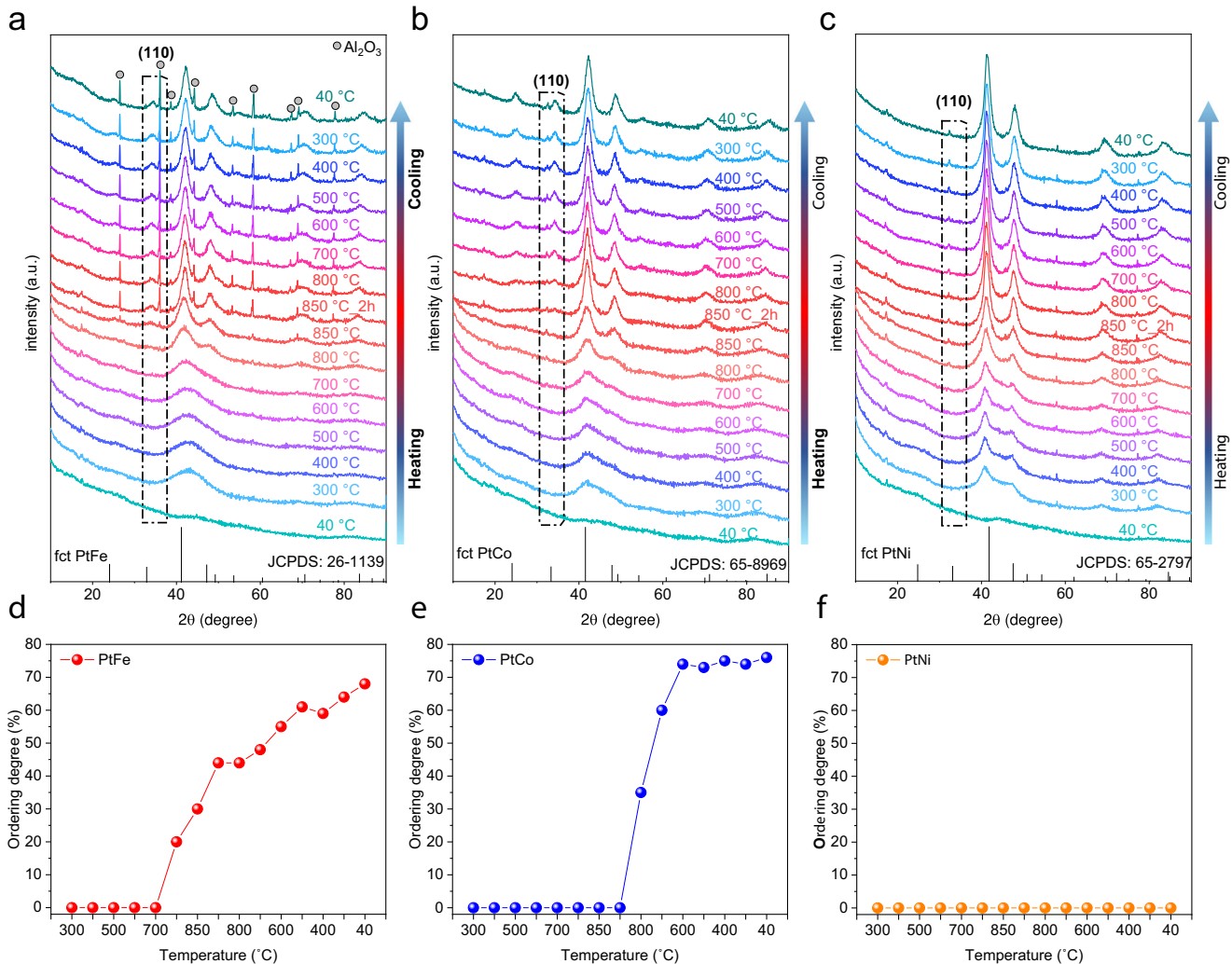

**Fig. 2 | In situ high-temperature X-ray diffraction (HT-XRD) results. a–c** In situ HT-XRD results of the Pt–Fe, Pt–Co, and Pt–Ni catalysts, respectively. The intensity of the characteristic superlattice peak of (110) reflects the evolution of ordered fct IMCs. The standard peaks for ordered fct PtFe (JCPDS no. 26-1139), PtCo (JCPDS no. 65-8969), and PtNi (JCPDS no. 65-2797) are also shown. As the experiment proceeded, the precursor sheet shrank to a smaller size, resulting in the emergence of characteristic peaks of $Al_2O_3$ substrates. **d–f** Calculated ordering degrees of PtFe, PtCo, and PtNi during the in-situ HT-XRD experiment, respectively.

ordered intermetallic structures concurrently via a thermodynamically driven phase transition. By contrast, as $T_{PT}$ of the PtCo (~830 °C) was much lower than that of the PtFe, the intermetallic PtCo structure was unfavorable at the annealing temperature of 850 °C. In this case, the disorder-to-order transition could be realized only when the sample was cooled to a lower temperature below $T_{PT}$ to enlarge the thermodynamic driving force for the phase transition. The absence of a disorder-to-order transition for the PtNi in the HT-XRD experiments was related to its significantly low $T_{PT}$ of ~630 °C, because the low annealing temperature below $T_{PT}$ was too low to overcome the kinetic energy barriers of atom ordering.

## Synthesis of highly ordered intermetallic compound (IMC) catalysts

On the basis of the above understanding of the $T_{PT}$-dependent structural evolution of PtFe, PtCo, and PtNi, we accordingly optimized the synthesis parameters for each case to achieve highly ordered IMC catalysts with a nominal total metal content of ~15 wt% with the BP2000 carbon black supports. The crystal sizes and ordering degrees of the catalysts in all syntheses are summarized in Supplementary Table 1. We first prepared the PtFe catalyst (denoted as PtFe-T900-2h) by the same impregnation with a similar annealing program as that

used for the HT-XRD experiment, involving the heating (5 °C min⁻¹), high-temperature holding (900 °C for 2 h), and cooling stages (-9 °C min⁻¹) (Fig. 3a). The ordering degree of PtFe-T900-2h (22%) was, however, much lower than that of the sample obtained in the HT-XRD (68%) (Fig. 2d). We surmised that the possible reason was the relatively longer heating stage in the HT-XRD, which would facilitate the alloying, or the longer cooling stage, and would be beneficial for the atom ordering. We then updated the synthesis protocol of PtFe by additionally holding the sample at a low temperature of 600 °C for 2 h to boost the atom ordering. The prepared PtFe-T900-2h-T600-6h showed an almost unchanged crystal size but increased ordering degree (46%). We also optimized the synthesis by increasing the annealing temperature to 1000 °C to prepare the PtFe-T1000-2h or prolonging the high-temperature holding temperature to 6 h for preparing PtFe-T900-6h to promote the alloying. A high ordering of 56% was realized for the PtFe-T1000-2h, but the crystal size increased significantly to 4.1 nm. Therefore, PtFe-T900-6h represented the optimal sample that exhibited the highest ordering degree of 61% and a moderate crystal size of 3.1 nm.

For the synthesis of PtCo with a relatively low $T_{PT}$, we first attempted low-temperature one-step annealing at 600 °C with a prolonged holding time of 6 h to prepare PtCo-T600-6h, but this

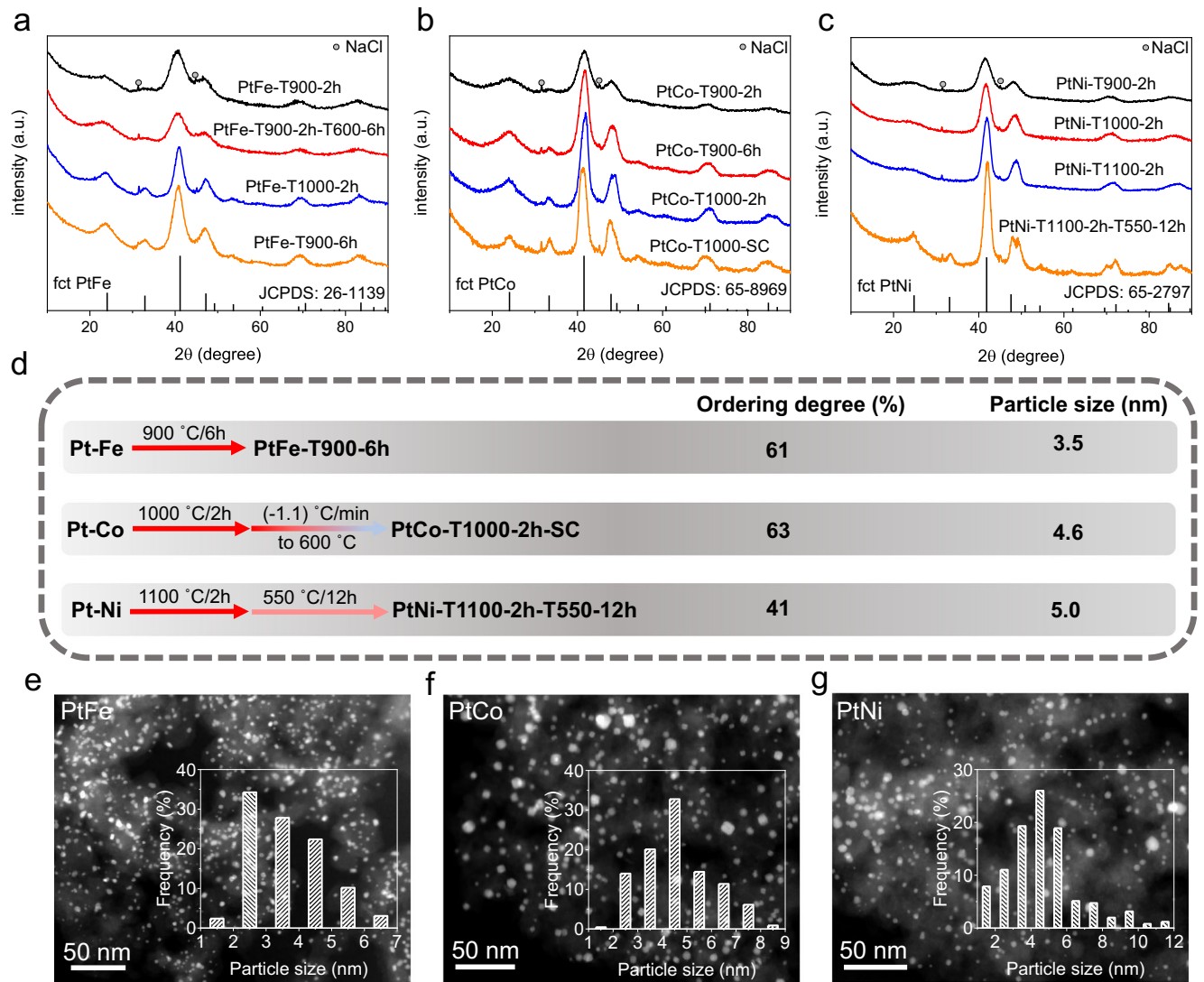

**Fig. 3 | Synthesis and structural characterization results. a–c** X-ray diffraction (XRD) patterns of the Pt–Fe, Pt–Co, and Pt–Ni catalysts, respectively. **d** Summary of the annealing protocols, ordering degrees, and average particle sizes of PtFe-T900-6h, PtCo-T1000-2h-SC, and PtNi-T1100-2h-T550-12h. **e–g** High-angular annular dark-field scanning transmission electron microscopy (HAADF-STEM) images of PtFe-T900-6h, PtCo-T1000-2h-SC, and PtNi-T1100-2h-T550-12h, respectively. The inserted histograms are the statistics of the particle size distributions of the corresponding samples.

approach failed to achieve intermetallic structures (Supplementary Fig. 2a). This result revealed that annealing at high temperatures above $T_{PT}$ was necessary for alloying Pt with a sufficient amount of Co to form disordered structures with a target stoichiometric ratio. The sample (denoted as PtCo-T900-2h) that was prepared by one-step annealing at a higher temperature of 900 °C for 2 h showed a low ordering degree of 15% (Fig. 3b). Either prolonging the holding time to 6 h (denoted as PtCo-T900-6h, with an ordering degree of 28%) or increasing the annealing temperature to 1000 °C (denoted as PtCo-T1000-2h, with an ordering degree of 37%) could promote the ordering degree. Because all three samples experienced exactly the same cooling step below $T_{PT}$ of PtCo for atom ordering, we ascribed the promoted ordering degrees of PtCo-T900-6h and PtCo-T1000-2h relative to that of PtCo-T900-2h to the improved alloying degree of the former two samples. To further promote the ordering degree, we finally adopted a separate alloying/ordering protocol that involved a high-temperature annealing step at 1000 °C for 2 h for alloying, followed by a very slow cooling step to 600 °C (−1.1 °C min$^{-1}$) for ordering to prepare a highly ordered fct PtCo catalyst (denoted as PtCo-T1000-2h-SC, with an ordering degree of 63%).

To date, there have been only a few reports on the synthesis of fct PtNi catalysts, which is likely because of the much lower $T_{PT}$ of PtNi than those of PtFe and PtCo. The abovementioned HT-XRD studies demonstrated that the intermetallic fct PtNi structure could not be generated during the conventional annealing process. We also failed to obtain fct PtNi catalysts by the conventional one-step impregnation/annealing synthesis at 900, 1000, and 1100 °C for 2 h (Fig. 3c). It is worth noting that the peak position of (111) shifted to a high 2θ degree with the increase in the annealing temperature. By comparison with the standard PDF card (JCPDS:65-2797), it was concluded that the ratio of Ni/Pt in PtNi-T1100-2h approached the target stoichiometric ratio of fct PtNi. We next additionally adopted a low-temperature-annealing holding stage at 600 °C for 6 h as the ordering step and followed a high-temperature-annealing holding stage at 1100 °C for alloying (Supplementary Fig. 2b). Fortunately, we observed weak superlattice peaks corresponding to fct PtNi for the prepared PtNi-T1100-2h-T600-6h, in spite of a low ordering degree of only 16%. The low ordering degree was likely caused by the temperature of the ordering step (~600 °C) being too close to $T_{PT}$ of PtNi (~630 °C), which would lead to a weak thermodynamic driving force for the disorder-to-order phase

transition, even for the PtNi alloy with the target stoichiometric ratio of 1:1. Accordingly, we then slightly decreased the temperature of the ordering step from 600 to 550 °C, and the ordering degree of the prepared PtNi-T1100-2h-T550-6h catalyst was increased to 28% (Supplementary Fig. 2b). We finally prolonged the annealing time of the ordering step at 550 °C to 12 h to obtain the optimal sample of PtNi-T1100-2h-T550-12h with a greatly promoted ordering degree of 41%.

In short, according to the abovementioned systematic optimization, we realized the synthesis of highly ordered PtFe-T900-6h, PtCo-T1000-2h-SC, and PtNi-T1100-2h-T550-12h catalysts by the industrially relevant impregnation method (Fig. 3d) (hereafter, denoted as PtFe, PtCo, and PtNi for short, respectively). We used high-angle annular dark-field scanning transmission electron microscopy (HAADF-STEM) to analyze the particle size distribution of these optimal catalysts, showing average particle sizes of 3.5, 4.6, and 5.0 nm for PtFe, PtCo, and PtNi, respectively (Fig. 3e–g), which were highly consistent with the values calculated by the Debye–Scherer equation based on the full-width at half-maximum of XRD patterns (Supplementary Table 1). Furthermore, aberration-corrected atomic-number($Z$)-contrast HAADF-STEM was conducted to analyze the crystal structures of the intermetallic catalysts on the atomic scale (Fig. 4a–c). For these three catalysts, the HAADF-STEM images along the [**110**] direction showed the alternating arrangement of Pt and Fe/Co/Ni atom columns, represented by brighter dots and darker dots, respectively (Pt columns had a higher intensity than Fe/Co/Ni columns with lower $Z$ values), indicating the presence of the L1$_0$-type fct intermetallic structure. Fast Fourier transform (FFT) patterns of the corresponding atomic-resolution HAADF-STEM images further verified the presence of fct intermetallic structures. Energy-dispersive X-ray spectroscopy (EDS) elemental mapping confirmed the homogeneous distributions of Pt and Fe/Co/Ni in the individual particles.

## Electrochemical performance

Prior to the electrochemical tests, the as-prepared optimal catalysts underwent acid leaching and low-temperature H$_2$ annealing (400 °C, 2 h) treatments successively to form an electrochemically stable and active core/shell structures consisting of an intermetallic Pt–M core and two to three atomic layers of Pt shells[6,17,30,31]. Inevitably, the ordering degrees of these treated catalysts decrease compared to the pristine ones because of the loss of transition metal atoms from the surface of the catalysts (Supplementary Fig. 3). We further noted that the ordering degree decline for PtFe was more severe than that for PtCo and PtNi upon the treatments, which was associated to the size effect. ICP-AES measurements confirmed that the loss of Fe from the PtFe catalyst with smaller average particle size was much higher than that of PtCo and PtNi catalysts with larger particle sizes (Supplementary Table 2).

Rotating disk electrode (RDE) tests were first conducted in an O$_2$-saturated 0.1 M HClO$_4$ solution at room temperature to evaluate the ORR performance of the treated PtFe, PtCo, PtNi, and commercial Pt/C catalysts. The PtFe catalyst exhibited the highest ORR activity among the three intermetallic catalysts, with a large half-wave potential ($E_{1/2}$) of 0.936 V, whereas the values were 0.926, 0.900, and 0.883 V for PtCo, PtNi, and Pt/C (Fig. 5a), respectively. The electrochemical surface areas (ECSA) of the catalysts were measured by the CO-stripping voltammetry (Supplementary Fig. 5). The PtFe catalyst showed a high ECSA of 85.4 m$^2$ g$^{-1}$, which was slightly lower than that of Pt/C (111.6 m$^2$ g$^{-1}$) (Supplementary Table 2). In terms of the mass activity (MA) and specific activity (SA), PtFe exhibited a high MA and SA of 2.61 A mg$^{-1}$ and 3.01 mA cm$^{-2}$ at 0.9 V, which were comparable to those of PtCo (MA, 2.10 A mg$_{Pt}^{-1}$; SA, 3.87 mA cm$^{-2}$) and higher than those of PtNi (MA, 0.53 A mg$_{Pt}^{-1}$; SA, 2.05 mA cm$^{-2}$) and Pt/C (MA, 0.34 A mg$_{Pt}^{-1}$; SA, 0.30 mA cm$^{-2}$) (Fig. 5b). Electrochemical impedance spectroscopy (EIS) analyses showed a lower charge transfer resistance in PtFe catalyst, confirming its superior activity (Supplementary Fig. 6). We attributed the highest MA of PtFe to its higher ECSA and ordering degree compared with those of PtCo and PtNi. The durability of the PtFe and Pt/C catalysts was evaluated by conducting an accelerated durability test (ADT) via the RDE. The PtFe catalyst exhibited a slight downshift of 9 mV for $E_{1/2}$ after 30,000 ADT cycles (Fig. 5c). The MA and SA of the PtFe maintained 70% and 78% of the initial values after 30,000 ADT cycles, respectively (Fig. 5d). By contrast, the Pt/C catalyst

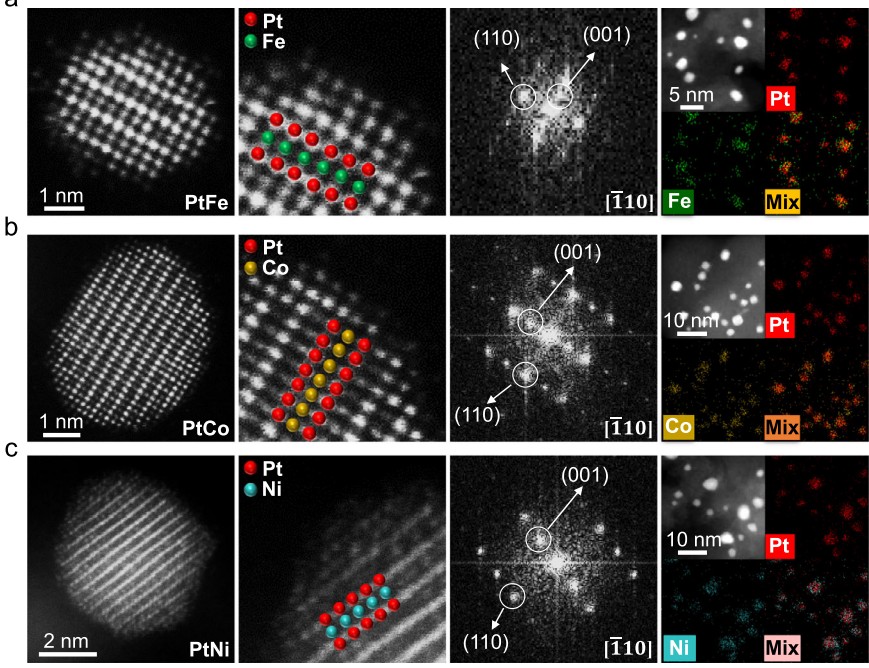

**Fig. 4 | Atomic scale HAADF-STEM characterization.** Atomic-resolution HAADF-STEM images, fast Fourier transform (FFT) patterns, and energy dispersive X-ray spectroscopy (EDS) elemental mappings of the PtFe (**a**), PtCo (**b**), and PtNi (**c**) IMC catalysts.

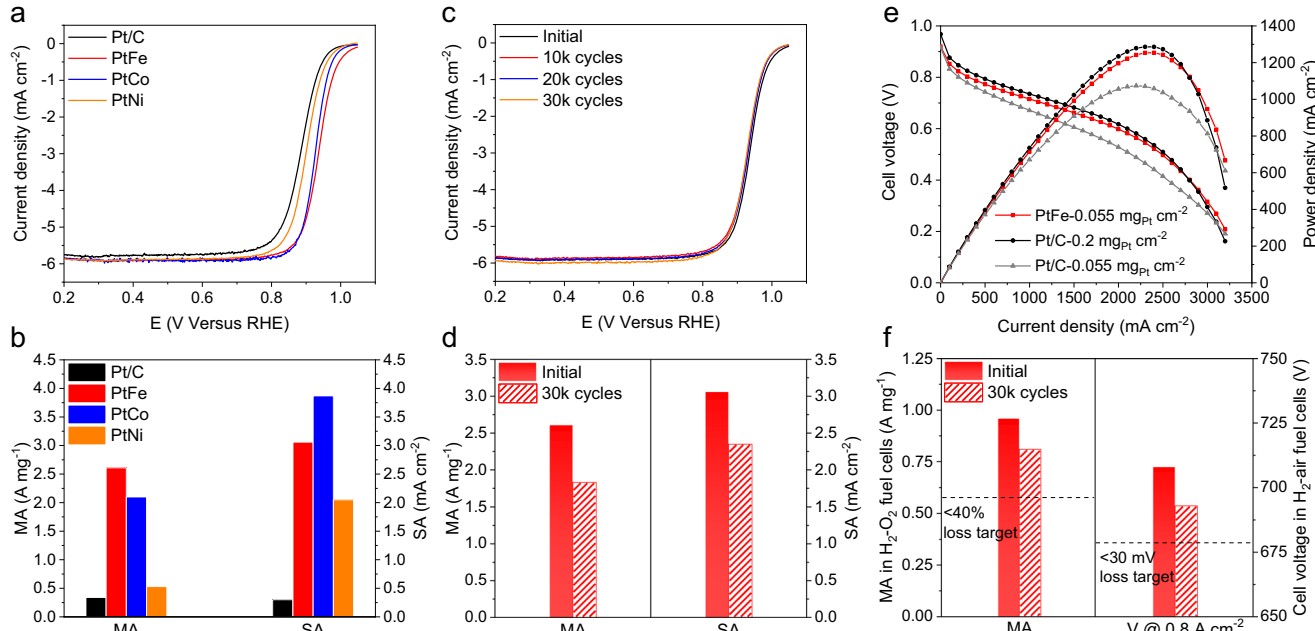

**Fig. 5 | Electrocatalytic performance. a** Oxygen reduction reaction (ORR) polarization curves of the Pt/C, PtFe, PtCo, and PtNi catalysts (with internal resistance corrected). **b** Comparison of mass activities (MAs) and specific activities (SAs) of the catalysts at 0.9 V (versus reversible hydrogen electrode (RHE)). The SAs were normalized by the electrochemical surface areas (ECSAs) estimated from CO stripping. **c** ORR polarization curves of the PtFe catalyst before and after accelerated durability test (ADT). **d** MAs and SAs of PtFe catalysts before and after ADT. **e** $H_2$–air single-cell polarization curves of PtFe and Pt/C cathodes. **f** MA in $H_2$–$O_2$ test and voltage at 0.8 A cm$^{-2}$ in $H_2$–air test of PtFe cathode before and after ADT.

showed a 50% drop of the MA after the ADT (Supplementary Fig. 8 and Supplementary Table 4).

We further performed membrane electrode assembly (MEA) tests to evaluate the catalyst performances in practical PEMFCs. The MAs of the catalysts were first evaluated at 0.9 $V_{iR\text{-correct}}$ in $H_2$–$O_2$ cell tests at 80 °C. The MA of the PtFe reached 0.96 A mg$^{-1}$, which was higher than that of Pt/C (0.20 A mg$^{-1}$). For $H_2$–air single-cell tests, the PtFe cathode exhibited a current density of 318 mA cm$^{-2}$ at 0.8 V in the kinetic region (Fig. 5e), which was higher than that of the Pt/C cathode (205 mA cm$^{-2}$) with the same loading. In addition, The PtFe cathode with a low-Pt loading of 0.055 mg$_{Pt}$ cm$^{-2}$ exhibited a comparable power density to that of the Pt/C cathode with a high Pt loading of 0.20 mg$_{Pt}$ cm$^{-2}$ in the high current density region, at which the MEA with low-Pt loading suffered from a higher local-$O_2$ transfer resistance[32]. We further evaluated the durability of PtFe by applying 30,000 cycles of square waves from 0.6 to 0.95 V in MEA. Notably, the PtFe catalyst retained 84% of its MA in $H_2$–$O_2$ tests and showed a voltage loss of less than 15 mV at 0.8 A cm$^{-2}$ in $H_2$–air tests after ADT (Fig. 5f and Supplementary Fig. 9). ICP-AES measurement was performed to evaluate the Fe loss during the ADT. The Fe/Pt ratio declined slightly from 0.41 o 0.36 after the ADT, indicated the high structural stability of the intermetallic catalysts[5].

## Discussion

In summary, we performed in situ HT-XRD to investigate the structural evolution during the high-temperature annealing synthesis of carbon-supported PtFe, PtCo, and PtNi intermetallic fuel cell catalysts. We identified the $T_{PT}$-dependent evolution process for PtFe, PtCo, and PtNi: the alloying and ordering proceeded concurrently in the high-temperature heating stage for PtFe with a high $T_{PT}$, while the alloying and ordering were separate for PtCo and PtNi with low $T_{PT}$ values. We accordingly designed a rational annealing process that involved separate high-temperature alloying and low-temperature ordering stages and thus realized the synthesis of highly ordered intermetallic catalysts with small particle size. The prepared highly ordered PtFe

intermetallic catalysts displayed a high ORR MA of 0.96 A mg$_{Pt}$ in $H_2$–$O_2$ fuel cells and an outstanding durability. Our results highlight the importance of understanding the alloying/ordering stages in maximizing the ordering degree of supported intermetallic catalysts with acceptable particle sizes for practical fuel cell applications.

## Methods

### In situ HT-XRD studies

To prepare the catalyst precursors for in situ HT-XRD studies, 50 mg BP2000, 14.8 mg $H_2PtCl_6$·$6H_2O$ (Sinopharm Chemical Reagent Co. LTD), and 8.5 mg $FeCl_3$·$6H_2O$ (Sinopharm Chemical Reagent Co. LTD) were added to 50 mL ethanol solution in round-bottom flask and then blended thoroughly by ultrasonic and stirring treatment overnight. The dried powder was obtained by using a rotary evaporator, followed by pressing in to a disk for in situ HT-XRD studies. A Panalytical Empyrean-diffractometer with a high-temperature reaction cell (XRK 900, Anton Paar GmbH) was used to perform the in situ HT-XRD studies. The annealing program in HT-XRD was shown in Supplementary Fig. 1a. The precursor disk was transferred to the sample room and subjected to annealing at 850 °C for 2 h (10 °C min$^{-1}$) and then cool down to 300 °C (10 °C min$^{-1}$) in 5 vol% $H_2/N_2$, followed by natural cooling to room temperature. The process for the preparation of the PtCo and PtNi precursor powders was similar to PtFe, by replacing $FeCl_3$·$6H_2O$ with $Co(NO_3)_2$·$6H_2O$ (10.8 mg) and $NiCl_2$·$6H_2O$ (10.2 mg), respectively.

### Synthesis of highly ordered PtFe, PtCo, and PtNi catalysts

The precursors were prepared in the same way with in situ HT-XRD studies. The dried powder was subjected to one-step or two-step annealing treatments at target temperature (5 °C min$^{-1}$) in 5 vol% $H_2/Ar$. For the synthesis of the optimal PtFe-T900-6h catalyst, a one-step annealing protocol with a prolonged high-temperature holding time was employed, in which the precursor was transferred to a tube furnace, heated at 900 °C (5 °C min$^{-1}$) for 6 h under flowing 5 vol% $H_2/Ar$, and cooled naturally to room temperature. For the synthesis of the

optimal PtCo-T1000-2h-SC catalyst, a one-step annealing protocol with very slowly cooling rate was employed, in which the precursor was heated at 1000 °C (5 °C min⁻¹) for 2 h in 5 vol% $H_2$/Ar and cooled down to 600 °C with −1.1 °C min⁻¹, followed with cooling naturally to room temperature. For the synthesis of the optimal PtNi-T1100-2h-T550-12h catalyst, a two-step annealing protocol was employed, in which the precursor was first heated at 1100 °C (5 °C min⁻¹) for 2 h and then heated at 550 °C for 12 h in 5 vol% $H_2$/Ar, followed with cooling naturally to room temperature. And the annealing program of other samples was listed in Supplementary Table 1.

## RDE tests

Before the electrochemical tests, all the three optimal catalysts including PtFe-T900-6h, PtCo-T1000-2h-SC, and PtNi-T1100-2h-T550-12h, were subjected to acid leaching (0.1 M $HClO_4$, 60 °C, 6 h), followed by annealing at 400 °C for 2 h in 5% $H_2$/Ar to form an active and stable core/shell catalysts. The inductively coupled plasma atomic emission spectrometry (ICP-AES) was performed to quantify the Pt contents in the final core/shell catalysts. CHI 760E electrochemical workstation and RDE were used to conduct electrochemical tests, which the RDE was composed of glass carbon as the working electrode. The counter electrode and reference electrode were Pt foil and $Hg/Hg_2SO_4$, respectively. And the $Hg/Hg_2SO_4$ was calibrated in $H_2$-saturated 0.1 M $HClO_4$. To prepare the catalyst ink, 4.0 mg catalysts, 2.0 mL iso-propanol (Sinopharm Chemical Reagent Co. LTD) and 40 μL 5 wt% Nafion solution (Sigma-Aldrich) were added into a bottle, followed by ultrasonically mixing for 30–60 min. A certain amount of catalysts ink (10.2 $μg_{Pt}$ cm⁻²) was deposited on the working electrode and dried under ambient temperature. The commercial Pt/C (TEC10E20E) was treated in the same way to compare the electrochemical performance. Before Linear sweep voltammetry (LSV) measurement, a certain cycle of cyclic voltammetry was performed. LSV was performed to evaluated the electrochemical performance in $O_2$-saturated 0.1 M $HClO_4$ at 10 mV s⁻¹ with 1600 rpm. The final performance was obtained after the correction of the capacitance current and solution resistance. The capacitance current was obtained in the same process excepting for $N_2$-saturated solution. The charge transfer resistance was obtained by electrochemical impedance spectroscopy measurement (BioLogic SP-150 electrochemical workstation). The ECSA was determined by integrating the CO stripping curves. The ADT was performed in $N_2$-saturated 0.1 M $HClO_4$ by cycling between 0.6 and 0.95 V (vs. RHE) at 100 mV s⁻¹ for 30,000 cycles.

## PEMFC tests

To prepare the cathode catalyst ink, the catalysts was ultrasonically dispersed in 1:1 n-propanol/water solution with 5 wt% Nafion solution for 30–60 min. The catalyst-coated-membrane (CCM) was prepared by using the ultrasonic spray (Sonotek) to direct spraying of the ink on Nafion membrane with an active area of 5 cm². The Pt loading of treated PtFe-T900-6h cathode and Pt/C anode were controlled to be 0.055 $mg_{Pt}$ cm⁻² and 0.025 $mg_{Pt}$ cm⁻², respectively. For comparison, the Pt/C cathodes with 0.20 and 0.055 $mg_{Pt}$ cm⁻² were prepared in the same process. The graphite plates were composed of 7 parallel flow channels with a depth of 0.8 mm and land width of 0.5 mm. MEA performance was evaluated in the Scribner 850e with Scribner 885 potentiostat. The $H_2$-$O_2$ MA was evaluated in constant voltage 0.9 $V_{iR-correct}$ for 15 min, which the MA was the average of current density in the last minute. The $H_2$−$O_2$ test condition was set at 80 °C, 100% RH, 150 $kPa_{abs}$ $H_2$/$O_2$ with 0.2/0.5 L min⁻¹. The $H_2$–air test conditions were set at two different conditions. One was set at 80 °C, 100% RH, 150 $kPa_{abs}$ $H_2$/air with 0.5/2 L min⁻¹, and another one was set at 94 °C, 100% RH, 250 $kPa_{abs}$ $H_2$/air with 0.5/2 L min⁻¹. The pressure drop between the inlet and outlet of flow field under the high gas flow rate condition was measured to be less than 10 kPa. The ADT was performed by square

wave cycling between 0.6 and 0.95 V for 30,000 cycles at 80 °C, 100% RH, atmospheric pressure $H_2$/$N_2$ with 0.2/0.075 L min⁻¹.

## Characterization

The ex situ XRD patterns were collected on Japan Rigaku DMax-γ A rotation anode X-ray diffractometer. HAADF-STEM and Aberration-corrected HAADF-STEM were conducted on FEI Talos F200X and JEM ARM200F TEM. EDS mapping was also conducted on FEI Talos F200X equipped with Super X-EDS system. The Optima 7300 DV was used to perform the ICP-AES measurements. XPS measurements were carried out on a VG ESCALAB MK II X-ray photoelectron spectrometer with an exciting source of Mg Kα = 1253.6 eV.

## Data availability

All data presented in this study are available from the corresponding authors (H.-W.L.) upon request.

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

## Acknowledgements
We acknowledge the funding support from the National Key Research and Development Program of China (Grant 2018YFA0702001), the National Natural Science Foundation of China (Grants 22071225 and 22221003), the Plan for Anhui Major Provincial Science & Technology Project (Grant 202203a0520013 and 202103a05020015), the Fundamental Research Funds for the Central Universities (Grant WK2060190103), the Joint Funds from Hefei National Synchrotron Radiation Laboratory (Grant KY2060000175), and Collaborative Innovation Program of Hefei Science Center of CAS (Grant 2021HSC-CIP015).

## Author contributions
H.-W.L. and W.-J.Z. conceived and designed the project. W.-J.Z., Q.-Q. Y., and P.Y. synthesized and characterized the catalysts. W.-J.Z. and C.W. performed the HT-XRD tests. W.-J.Z., L.T., and H.-W.L. co-wrote the manuscript. All authors discussed the results and commented on the manuscript.

## Competing interests
The authors declare no competing interests.
