## [Peer review file · Nature Communications]

REVIEWER COMMENTS

Reviewer #1 (Remarks to the Author):

In this work, the authors develop crystal growth and annealing strategies for growing ordered intermetallic Pt alloy electrocatalysts, and then evaluate the structure and performance of these materials with conventional techniques. What sets this work apart from previous studies is the use of the in situ XRD during catalyst synthesis. These data allow the team to rationally design the temperature treatment to grow catalysts with the desired properties. This is important but absent from previous literature, where annealing conditions are frustratingly selected often at random, after screening conditions which are not presented.

While there does not seem to be any large discoveries or exceptionally active catalysts disclosed in this manuscript, this is exactly the kind of high quality work and well-presented paper that I would read, cite, and use in my own research. In particular, I enjoyed how the manuscript concisely presents the key results, without diluting the message with excessive supplementary characterization (XPS, EXAFS, DFT, etc). There are a number of tricky aspects to electrocatalysis work (eg normalizing alloy surface areas with CO stripping voltammetry) which are often neglected but which these authors handle flawlessly, and which builds trust with the reader. For this reason I think this study would have relatively high impact.

I recommend publication of this manuscript following some revisions, in roughly descending order of importance:

1. Did you measure XRD on the intermetallic MEA after testing? What influence does the leaching and H₂ annealing steps have on the morphology and intermetallic structure, especially the ordering degree by XRD? If you see the superlattice reflections you will know whether or not the intermetallic structure actually survives the fuel cell testing. This is presumably quite important for understanding the stability of these catalysts. I would not trust post-mortem TEM for this, since statistical sampling is critically important here.

2. There are some issues with the PEMFC testing. Was this performed on a 5cm² cell? The flow rates reported are quite high. If my math is correct, for 5cm², 2L/min air at 2.5 bar provides a stoichiometry of over 50 at maximum power. Typical values for actual cells are 2-3. This artificially enhances maximum power density achieved at very low loadings by removing water from the cell, and the performance is mostly a function of catalyst layer transport phenomena, not superior electrocatalyst performance. For catalyst studies it is totally acceptable to use high flow rates to remove mass transport considerations, but if this is done then it is not fair to highlight excellent power densities and to compare results with DOE targets for actual devices running at stoichiometric flow. To compare catalysts, it is best to look at

the low power density region where voltage losses are from electrochemical kinetics, as performed in your reference 5 (DOI: 10.1016/j.joule.2018.09.016). I understand there are many literature examples which use outrageous flow rates, O₂ instead of air, and high backpressure to achieve extreme power densities at low loadings, but this is not acceptable practice.

3. For the in situ XRD, using the peak intensity ratio method is not ideal for these systems. Rietveld analysis is preferred but the data quality might not be sufficient. Partially ordered intermetallics can have unpredictable intensities because the scattering power of Pt and Co/Fe/Ni are very different. Pt or M occupancies above or below 50% in either slab will dramatically skew the results. It would be nice to tabulate lattice parameters in the SI, especially since there is some discussion about peak shifts. The catalyst particle sizes determined by XRD should be corrected for instrumental broadening, but this may be negligible for such small particles depending on your instrument.

4. In the introduction, there are a few broad and controversial claims which are referenced but that I don't necessarily agree with. I suggest toning down the strength of these claims, since there is quite a bit of unresolved and interesting fundamental questions in this emerging field. The claim that ordered catalysts are distinctly advantageous over disordered alloys is not really supported by the broader literature, which is also why these catalysts are not used commercially. The reasons for any discrepancy in activity between ordered and disordered catalysts also remains unclear, since a dealloyed shell is quickly developed, and strain/ligand effects do not normally extend past 2-3 monolayers of Pt shell. This is especially true for the L10 PtM phases, which are less Pt rich than typical alloys with 25% base metal content, and therefore dealloy more severely. The idea that ordered intermetallics are universally more stable is also questionable, and there is not to my knowledge a straightforward relationship between leaching rate and formation enthalpy for alloys with different unit cells (ie intermetallic vs solid solution).

5. The ICP results are interesting, since they let you confirm the maximum possible ordering. Could you please tabulate the Pt-M ratios in the SI? Wt% is not helpful here. It would be very useful to have this data for the as-fabricated material, after leaching, and after electrochemical cell testing. Please also specify if the results in Table S2 are from before or after leaching.

6. The writing quality is generally quite good, there are a few errors that should be proofread ideally by a native English speaker before publication. For example donated vs denoted, comprising vs comparing(?)

7. Why does the diffusion limited current change in Fig S5? Is the catalyst delaminating from the RDE?

8. What conversion factor was used to calculate the surface area from the integrated CO stripping charge?

Reviewer #2 (Remarks to the Author):

Nice and interesting manuscript on PtFe electrocatalysts for ORR, tested in PEMFC with H₂/O₂ and H₂/air. Very good performance, and durability. Very low Pt content.

The manuscript can be accepted. I only recommend to benchmark the catalysts with literature data (there are papers with PtFe at the cathode side), and to verify the performance of these catalysts according to the DOE targets, also considering the amount of Pt.

PGM-free activity* > 44 mA/cm² @ 900 mV IR-free (PGM total content on both electrodes of 125 g/kW at 150 kPa,abs)

Loss in performance at 0.8 A/cm² < 30 mV

PGM total loading (anode side) < 0.1 mg(PGM)/cm²

Mass activity > 440 mA/mg(PGM)₁ @ 900 mV IR-free

Loss in initial catalytic activity < 40 % mass activity loss

See these papers as a reference:

Thompson ST, et al, ElectroCat: DOE's approach to PGM-free catalyst and electrode R&D. Solid State Ionics 319 (2018) 68–76. <https://doi.org/10.1016/j.ssi.2018.01.030>

Brouzgou A et al, Low and non-platinum electrocatalysts for PEMFCs: Current status, challenges and prospects. Appl Catal B Environ 127 (2012) 371–388. <https://doi.org/10.1016/j.apcatb.2012.08.031>

Reviewer #3 (Remarks to the Author):

This manuscript reports on “Phase diagrams guide synthesis of highly ordered intermetallic electrocatalysts: separating alloying and ordering stages”. Enhancing the electrocatalytic activity of the

commercial available Pt/C catalysts toward electrochemical-assisted ORR reaction as well as reducing the cost of the PEMFCs stacks remain as challenging milestone on the way of PEMFCs commercialization. Thus, it is of great importance to develop highly active yet cost-effective electrode materials, especially as the cathode-side of the PEMFCs where the sluggish oxygen reduction reaction occurs. The present paper somewhat addresses the above mentioned topics. However, there is more characteristics and electrochemical information needed to convince readers that the proposed electrode materials can be practically used in real practical applications. The manuscript can be publishable after the revision as described below:

1. The post characterization analyses including XRD and ICP should be done to show any trace of secondary alloying metal leakage after Accelerated Stress Tests (ASTs) from the structure of the electrocatalysts.
2. The authors mentioned that they have used a mixture of 4.0 mg catalysts, 2.0 mL isopropanol and 40 μ L 5 wt% Nafion solution without using water. Since using the pure alcohol can deactivate the Pt, I would like to recommend the authors to use a mixture of water and IPA, as suggested by DOE.
3. In order to make a better comparison between the electrochemical behavior of the as-prepared electrocatalysts toward ORR in terms of charge transfer resistance, the authors are recommended to provide EIS analysis.
4. In order to show the effect of the alloying metal (M) on the d-band of the Pt, the authors should provide XPS results for all the electrode materials and compare it with the pure platinum.

Reviewer #1 :

In this work, the authors develop crystal growth and annealing strategies for growing ordered intermetallic Pt alloy electrocatalysts, and then evaluate the structure and performance of these materials with conventional techniques. What sets this work apart from previous studies is the use of the in situ XRD during catalyst synthesis. These data allow the team to rationally design the temperature treatment to grow catalysts with the desired properties. This is important but absent from previous literature, where annealing conditions are frustratingly selected often at random, after screening conditions which are not presented.

While there does not seem to be any large discoveries or exceptionally active catalysts disclosed in this manuscript, this is exactly the kind of high quality work and well-presented paper that I would read, cite, and use in my own research. In particular, I enjoyed how the manuscript concisely presents the key results, without diluting the message with excessive supplementary characterization (XPS, EXAFS, DFT, etc). There are a number of tricky aspects to electrocatalysis work (eg normalizing alloy surface areas with CO stripping voltammetry) which are often neglected but which these authors handle flawlessly, and which builds trust with the reader. For this reason I think this study would have relatively high impact.

Response: We appreciate the reviewer's encouraging comments here.

I recommend publication of this manuscript following some revisions, in roughly descending order of importance:

1. Did you measure XRD on the intermetallic MEA after testing? What influence does the leaching and H₂ annealing steps have on the morphology and intermetallic structure, especially the ordering degree by XRD? If you see the superlattice reflections you will know whether or not the intermetallic structure actually survives the fuel cell testing. This is presumably quite important for understanding the stability of these catalysts. I would not trust post-mortem TEM for this, since statistical sampling is critically important here.

Response: We appreciate the reviewer's helpful comments on the XRD data of the catalysts. According to the reviewer's suggestions here, we performed additional XRD measurement to analyze the structure change of our catalysts during 1) the acid leaching and H₂ annealing before MEA tests and 2) after the MEA durability test.

The XRD patterns of PtFe, PtCo, and PtNi catalysts after leaching and H₂ annealing treatments are shown in Fig. R1. Inevitably, the ordering degrees of these treated catalysts decrease compared to the pristine ones because of the loss of transition metal atoms from the surface of the catalysts. We further noted that the ordering degree decline for PtFe was more severe than that for PtCo and PtNi upon the treatments, which was associated to the size effect. ICP-AES measurements confirmed that the loss of Fe from the PtFe catalyst with smaller average particle size (3.5 nm) was much higher than that of PtCo (4.6 nm) and PtNi (5.0 nm) catalysts with larger particle sizes (supplementary Table S2).

We also tried to measure the XRD of the PtFe catalyst in MEA before and after ADT. We failed to peel off the cathode catalyst layer from the catalyst-coated-membrane (CCM) that was composed of the PtFe cathode catalyst layer, the Pt/C anode catalyst layer, and the membranes. The XRD patterns of blank membrane, PtFe-CCM-Initial, and PtFe-CCM-ADT are shown in Fig. R2. The PtFe-MEA-Initial was prepared by spraying the catalyst ink on the blank membrane without testing; and the PtFe-CCM-ADT was the CCM after ADT. Unfortunately, we could not recognize the superlattice peaks of PtFe even in the XRD patterns of PtFe-CCM-Initial, owing to 1) the interference of unknown peaks located at $\sim 33^\circ$ (2θ) of the black membranes and 2) the low signal intensity of XRD for the low catalyst-loading CCM. We further performed the ICP-AES measurements to analyze the possible Fe loss from the catalyst during the ADT. The Fe/Pt ratio only declined from 0.41 to 0.36 after ADT, indicated the satisfactory electrochemical stability of the PtFe catalysts.

Fig. R1. XRD patterns of the PtFe (a), PtCo (b), and PtNi (c) catalysts before and after leaching and H₂ annealing steps.

Fig. R2. XRD patterns of the blank membrane, PtFe-CCM-Initial, and PtFe-CCM-ADT.

2. There are some issues with the PEMFC testing. Was this performed on a 5cm² cell? The flow rates reported are quite high. If my math is correct, for 5cm², 2L/min air at 2.5 bar provides a stoichiometry of over 50 at maximum power. Typical values for actual cells are 2-3. This artificially enhances maximum power density achieved at very low loadings by removing water from the cell, and the performance is mostly a function of catalyst layer transport phenomena, not superior electrocatalyst performance. For catalyst studies it is totally acceptable to use high flow rates to remove mass transport

considerations, but if this is done then it is not fair to highlight excellent power densities and to compare results with DOE targets for actual devices running at stoichiometric flow. To compare catalysts, it is best to look at the low power density region where voltage losses are from electrochemical kinetics, as performed in your reference 5 (DOI: 10.1016/j.joule.2018.09.016). I understand there are many literature examples which use outrageous flow rates, O₂ instead of air, and high backpressure to achieve extreme power densities at low loadings, but this is not acceptable practice.

Response: We appreciate the reviewer's helpful comments on the PEMFC testing. The CCM was indeed prepared with an active area of 5 cm² on Nafion membrane. According to our calculation, the stoichiometry is of around 20 for 2 L min⁻¹ air at 2.5 bar at 12 A corresponding to the maximum power density, which is indeed far higher than the typical values for actual cells. According to the reviewer's comments, we have deleted the related discussion on the comparison of the power density with DOE targets and instead we move to the discussion on the current density at high voltages, which is more relevant to the electrochemical kinetics. This part has been reorganized in the revised manuscript as follows:

“For H₂-air single-cell tests, the PtFe cathode exhibited a current density of 318 mA cm⁻² at 0.8 V, which was higher than that of the Pt/C cathode (205 mA cm⁻²) with the same loading. In addition, the PtFe cathode with a low-Pt loading of 0.055 mg_{Pt} cm⁻² exhibited a comparable power density to that of the Pt/C cathode with a high Pt loading of 0.20 mg_{Pt} cm⁻² in the high current density region, at which the low-Pt loading cathode suffered from a higher local-O₂ transfer resistance¹. (Page 8)

The related discussion has been updated in the Abstract, Introduction, and the Discussion sections

3. For the in situ XRD, using the peak intensity ratio method is not ideal for these systems. Rietveld analysis is preferred but the data quality might not be sufficient. Partially ordered intermetallics can have unpredictable intensities because the scattering power of Pt and Co/Fe/Ni are very different. Pt or M occupancies above or below 50% in either slab will dramatically skew the results. It would be nice to tabulate lattice parameters in the SI, especially since there is some discussion about peak shifts. The catalyst particle sizes determined by XRD should be corrected for instrumental broadening, but this may be negligible for such small particles depending on your instrument.

Response: We appreciate the reviewer's helpful comments here. As reviewer commented, the data quality was indeed not sufficient for the Rietveld analysis. Although the accurate values of ordering degree could not be obtained, the increased trend of ordering degree between different annealing process was very clear from the current XRD data. As suggested by the reviewer, the values of interplanar spacing of (111) of the catalysts have been added in Supplementary Table 1, which were calculated by the Bragg's law based on the XRD results. We agree with the reviewer that the influence of instrumental broadening in XRD was negligible. Moreover, the XRD data actually reflects the crystal size instead of particle size. Therefore, we meanwhile discussed the average particle size based on the HADDF-STEM analyses.

4. In the introduction, there are a few broad and controversial claims which are referenced but that I don't necessarily agree with. I suggest toning down the strength of these claims, since there is quite a bit of unresolved and interesting fundamental questions in this emerging field. The claim that ordered catalysts are distinctly advantageous over disordered alloys is not really supported by the broader literature, which is also why these catalysts are not used commercially. The reasons for any discrepancy in activity between ordered and disordered catalysts also remains unclear, since a dealloyed shell is quickly developed, and strain/ligand effects do not normally extend past 2-3 monolayers of Pt shell. This is especially true for the L10 PtM phases, which are less Pt rich than typical alloys with 25% base metal content, and therefore dealloy more severely. The idea that ordered intermetallics are universally more stable is also questionable, and there is not to my knowledge a straightforward relationship between leaching rate and formation enthalpy for alloys with different unit cells (ie intermetallic vs solid solution).

Response: We appreciate the reviewer's constructive comments on the advantage of ordered intermetallic structures over disordered solid solution structures. We fully agree with the reviewer that so far there are no solid evidences that could strongly verify the advantage of ordered alloys over disordered ones for the improvement of ORR activity as well as the durability. According to the reviewer's suggestion here, we have reorganized the Introduction part of the revised manuscript by toning down the strength of the relevant claims. (Page 2)

5. The ICP results are interesting, since they let you confirm the maximum possible ordering. Could you please tabulate the Pt-M ratios in the SI? Wt% is not helpful here. It would be very useful to have this data for the as-fabricated material, after leaching, and after electrochemical cell testing. Please also specify if the results in Table S2 are from before or after leaching.

Response: We appreciate the reviewer's helpful comments here. The ICP-AES results of the as-fabricated catalyst and the catalysts after leaching have been added in the Supplementary Table S2, which indicated the maintained of the M was highly dependent on the particle size. Moreover, upon the MEA ADT, the Fe/Pt ratio in the PtFe catalysts declined slightly from 0.41 to 0.36, indicated the satisfactory electrochemical stability of the PtFe catalysts.

6. The writing quality is generally quite good, there are a few errors that should be proofread ideally by a native English speaker before publication. For example donated vs denoted, comprising vs comparing(?)

Response: We appreciate the reviewer's helpful comments here. We have employed a language service company (LetPub) to polish the English of the whole main text. All the changes were highlighted in red in the revised manuscript.

7. Why does the diffusion limited current change in Fig S5? Is the catalyst delaminating from the RDE?

Response: We appreciate the reviewer for noting this detail. It is possible that the catalysts delaminated from the glass carbon electrode. We retested the commercial Pt/C catalyst by RDE and the data have been updated in the revised manuscript and SI.

Fig. R3. (a) LSV, (b) CV, and (c) CO stripping curves of the Pt/C catalyst before and after 30,000 cycles ADT.

8. What conversion factor was used to calculate the surface area from the integrated CO stripping charge?

Response: The conversion factor of 0.42 mC cm⁻² for a monolayer of adsorbed CO was used to calculate the surface area from the integrated CO stripping charge.

Reviewer #2:

Nice and interesting manuscript on PtFe electrocatalysts for ORR, tested in PEMFC with H₂/O₂ and H₂/air. Very good performance, and durability. Very low Pt content.

Response: We appreciate the reviewer's encouraging comments here.

The manuscript can be accepted. I only recommend to benchmark the catalysts with literature data (there are papers with PtFe at the cathode side), and to verify the performance of these catalysts according to the DOE targets, also considering the amount of Pt.

PGM-free activity* > 44 mA/cm² @ 900 mV IR-free (PGM total content on both electrodes of 125 g/kW at 150 kPa,abs)

Loss in performance at 0.8 A/cm² < 30 mV

PGM total loading (anode side) < 0.1 mg(PGM)/cm²

Mass activity > 440 mA/mg(PGM)₁ @ 900 mV IR-free

Loss in initial catalytic activity < 40 % mass activity loss

See these papers as a reference:

Thompson ST, et al, ElectroCat: DOE's approach to PGM-free catalyst and electrode R&D. Solid State Ionics 319 (2018) 68–76. <https://doi.org/10.1016/j.ssi.2018.01.030>

Brouzgou A et al, Low and non-platinum electrocatalysts for PEMFCs: Current status, challenges and prospects. Appl Catal B Environ 127 (2012) 371–388.

<https://doi.org/10.1016/j.apcatb.2012.08.031>

Response: We appreciate the reviewer's comments here and we tried to tabulate the performance of the PtFe catalysts for the comparison with literature data (Table R1). Unfortunately, we found that the test conditions of the reported catalysts were vastly different to each other, including the type of cathode gas (air/O₂), size of MEA, gas flow, Pt loading, back pressure, relative humidity, flow field structure, etc. These experimental parameters have even more pronounced impact to the MEA performance than the catalysts, in particular for the high-current-density performance. Therefore, it is highly challenging to get reliable conclusion from such comparison. The mentioned DOE targets for the PGM catalysts are met well in our PtFe catalyst.

Table R1. Summary PEMFC performance of reported PtFe catalysts.

Catalyst	Pt loading in cathode (mg _{Pt} cm ⁻²)	Total Pt loading in MEA (mg _{Pt} cm ⁻²)	MA (A mg ⁻¹)	Maximum power density (mW cm ⁻²)	MA loss (%)	Voltage loss at 0.8 A cm ⁻² (mV)
PtFe This work	0.055	0.080✓	0.96✓	1260	16✓	15✓
PtFe-N-C ³	0.015	0.115	0.77✓	1080*	3✓	~0✓
FePt/VC ⁴	0.8	1.6	/	~110	/	21 mV at 0.2 A cm ⁻²
PtFe/C ⁵	0.8	1.1	0.049	~420*	/	/

PtFe-N-C ⁶	0.0067	0.1067	/	750*	/	/
cBCP-PtFe ⁷	0.01	0.21	0.81	0.96*	/	/
Pt ₃ Fe ₁ /NCB ⁸	0.3	0.5	/	1.08*	/	/
FePt ⁹	0.1	0.2	/	~550*	/	/
L1 ₀ - FePt/Pt ¹⁰	0.113	/	0.21	~600	✓	/
Pt-Fe ¹¹	0.32	0.54	/	-650*	/	/
fct-PtFe ¹²	0.2	0.4	/	1013*	/	✓
fct-PtFe/C ¹³	0.2	0.4	/	~500	/	/
PtFe/VC ¹⁴	0.4	0.2	/	~450*	/	/
PtFe/C-M ¹⁵	/	0.3		~250*		

The symbol of ✓ indicates the DOE targets are met.

MA: mass activity at 0.9 V in MEA.

~The power density of some studies is not given directly but is estimated from the polarization curve.

*The power density of some studies is estimated in the H₂-O₂ fuel cell tests.

Reviewer #3:

This manuscript reports on “Phase diagrams guide synthesis of highly ordered intermetallic electrocatalysts: separating alloying and ordering stages”. Enhancing the electrocatalytic activity of the commercial available Pt/C catalysts toward electrochemical-assisted ORR reaction as well as reducing the cost of the PEMFCs stacks remain as challenging milestone on the way of PEMFCs commercialization. Thus, it is of great importance to develop highly active yet cost-effective electrode materials, especially as the cathode-side of the PEMFCs where the sluggish oxygen reduction reaction occurs. The present paper somewhat addresses the above mentioned topics. However, there is more characteristics and electrochemical information needed to convince readers that the proposed electrode materials can be practically used in real practical applications. The manuscript can be publishable after the revision as described below:

Response: We appreciate the reviewer’s encouraging comments here.

1. The post characterization analyses including XRD and ICP should be done to show any trace of secondary alloying metal leakage after Accelerated Stress Tests (ASTs) from the structure of the electrocatalysts.

Response: We appreciate the reviewer’s helpful comments here. Accordingly, we tried to measure the XRD of the PtFe catalyst in MEA before and after ADT. We failed to peel off the cathode catalyst layer from the catalyst-coated-membrane (CCM) that was composed of the PtFe cathode catalyst layer, the Pt/C anode catalyst layer, and the membranes. The XRD patterns of blank membrane, PtFe-CCM-Initial, and PtFe-CCM-ADT are shown in Fig. R4. The PtFe-MEA-Initial was prepared by spraying the catalyst ink on the blank membrane without testing; and the PtFe-CCM-ADT was the CCM after ADT. Unfortunately, we could not recognize the superlattice peaks of PtFe even in the XRD patterns of PtFe-CCM-Initial, owing to 1) the interference of unknown peaks located at $\sim 33^\circ$ (2θ) and 2) the low signal intensity of XRD for the low catalyst-loading CCM. We further performed the ICP-AES measurements to analyze the possible Fe loss from the catalyst during the ADT. The Fe/Pt ratio only declined from 0.41 to 0.36 after ADT, indicated the satisfactory electrochemical stability of the PtFe catalysts.

The related discussions have been added in revised manuscript. (Page 8)

Fig. R4. (a) XRD pattern of PtFe catalyst after leaching and H₂ annealing steps. (b)–(c) XRD patterns of the blank membrane, PtFe-CCM-Initial, and PtFe-CCM-ADT, respectively.

2. The authors mentioned that they have used a mixture of 4.0 mg catalysts, 2.0 mL isopropanol and 40 μ L 5 wt% Nafion solution without using water. Since using the pure alcohol can deactivate the Pt, I would like to recommend the authors to use a mixture of water and IPA, as suggested by DOE.

Response: We appreciate the reviewer's helpful comments here. As we known, the RDE test is sensitive to the quality of film deposited on glass carbon electrode. We found that the using of pure IPA instead of a mixture of water and IPA could increase the quality of film, which was also frequently reported previously by other groups¹⁶⁻¹⁹. However, the ink used for MEA test was indeed prepared with a mixture of IPA and water (1:1), as suggested by the US DOE.

3. In order to make a better comparison between the electrochemical behavior of the as-prepared electrocatalysts toward ORR in terms of charge transfer resistance, the authors are recommended to provide EIS analysis.

Response: We appreciate the reviewer's useful comments here. The electrochemical impedance spectroscopy (EIS) measurements were performed at 0.9V (vs. RHE) with a frequency from 100, 000 to 0.01 Hz in O₂-saturated 0.1M HClO₄ solution with 1600 rpm. As shown in Fig. R5, the PtFe catalyst exhibited a lower charge transfer resistance, which was consistent well with the superior performance in RDE tests. The related discussions have been added in revised manuscript (Page 7) and SI.

Fig. R5. Nyquist plots tested at 0.9V (vs. RHE) of Pt/C, PtFe, PtCo, and PtNi catalysts.

4. In order to show the effect of the alloying metal (M) on the d-band of the Pt, the authors should provide XPS results for all the electrode materials and compare it with the pure platinum.

Response: According to the reviewer's comments here, we performed the X-ray photoelectron spectroscopy (XPS) analyses. We observed by XPS that the Pt 4f binding energy of the Pt-M catalyst shift negatively compared to that of Pt/C, indicating the charge donating from M to Pt in the alloy catalysts. The XPS data has been added in the SI.

Fig. R5. XPS spectra (Pt 4f) of the Pt/C catalyst and the PtFe, PtCo, and PtNi catalysts after leaching and H₂ annealing treatments.

Reference:

- 1 Tang, M., Zhang, S. & Chen, S. Pt utilization in proton exchange membrane fuel cells: structure impacting factors and mechanistic insights. *Chem. Soc. Rev.*, (2022).
- 2 Zhao, Z. *et al.* Tailoring a Three-Phase Microenvironment for High-Performance Oxygen Reduction Reaction in Proton Exchange Membrane Fuel Cells. *Matter* **3**, 1774-1790, (2020).
- 3 Xiao, F. *et al.* Atomically dispersed Pt and Fe sites and Pt-Fe nanoparticles for durable proton exchange membrane fuel cells. *Nat. Catal.* **5**, 503-512, (2022).
- 4 Sapkota, P., Lim, S. & Aguey-Zinsou, K.-F. Superior Performance of an Iron-Platinum/Vulcan Carbon Fuel Cell Catalyst. *Catalysts* **12**, (2022).
- 5 Mayorova, N. A. *et al.* Nanoscale catalyst based on a heterometallic carboxylate complex of platinum and iron for hydrogen-air fuel cells. *Mater. Chem. Phys.* **259**, (2021).
- 6 Xiao, F. *et al.* Durable hybrid electrocatalysts for proton exchange membrane fuel cells. *Nano Energy* **77**, (2020).
- 7 Choi, J. *et al.* Highly durable fuel cell catalysts using crosslinkable block copolymer-based carbon supports with ultralow Pt loadings. *Energy Environ. Sci.* **13**, 4921-4929, (2020).
- 8 Yang, H., Ko, Y., Lee, W., Züttel, A. & Kim, W. Nitrogen-doped carbon black supported Pt-M (M = Pd, Fe, Ni) alloy catalysts for oxygen reduction reaction in proton exchange membrane fuel cell. *Mater. Today Energy* **13**, 374-381, (2019).
- 9 Sandström, R., Hu, G. & Wågberg, T. Compositional Evaluation of Coreduced Fe-Pt Metal Acetylacetonates as PEM Fuel Cell Cathode Catalyst. *ACS Appl. Energy Mater.* **1**, 7106-7115, (2018).
- 10 Li, J. *et al.* Fe Stabilization by Intermetallic L10-FePt and Pt Catalysis Enhancement in L10-FePt/Pt Nanoparticles for Efficient Oxygen Reduction Reaction in Fuel Cells. *J. Am. Chem. Soc.* **140**, 2926-2932, (2018).
- 11 Tamaki, T. *et al.* Connected nanoparticle catalysts possessing a porous, hollow capsule structure as carbon-free electrocatalysts for oxygen reduction in polymer electrolyte fuel cells. *Energy Environ. Sci.* **8**, 3545-3549, (2015).
- 12 Lee, J. *et al.* Development of Highly Stable and Mass Transfer-Enhanced Cathode Catalysts: Support-Free Electrospun Intermetallic FePt Nanotubes for Polymer Electrolyte Membrane Fuel Cells. *Adv. Energy Mater.* **5**, (2015).
- 13 Chung, D. Y. *et al.* Highly Durable and Active PtFe Nanocatalyst for Electrochemical Oxygen Reduction Reaction. *J. Am. Chem. Soc.* **137**, 15478-15485, (2015).
- 14 Yang, D.-S., Kim, M.-S., Song, M. Y. & Yu, J.-S. Highly efficient supported PtFe cathode electrocatalysts prepared by homogeneous deposition for proton exchange membrane fuel cell. *Int. J. Hydrogen Energy* **37**, 13681-13688, (2012).
- 15 Xiong, L. & Manthiram, A. Nanostructured Pt-M/C (M=Fe and Co) catalysts prepared by a microemulsion method for oxygen reduction in proton exchange membrane fuel cells. *Electrochim. Acta* **50**, 2323-2329, (2005).
- 16 Chong, L. *et al.* Ultralow-loading platinum-cobalt fuel cell catalysts derived from imidazolate frameworks. *Science* **362**, 1276-1281, (2018).
- 17 Wang, D. *et al.* Structurally ordered intermetallic platinum-cobalt core-shell nanoparticles with enhanced activity and stability as oxygen reduction electrocatalysts. *Nat. Mater.* **12**, 81-87, (2013).
- 18 Xiong, Y. *et al.* Revealing the atomic ordering of binary intermetallics using in situ heating

techniques at multilength scales. *Proc. Natl Acad. Sci.* **116**, 1974-1983, (2019).

- 19 Chen, H. *et al.* A surfactant-free strategy for synthesizing and processing intermetallic platinum-based nanoparticle catalysts. *J. Am. Chem. Soc.* **134**, 18453-18459, (2012).

REVIEWERS' COMMENTS

Reviewer #1 (Remarks to the Author):

The authors have responded thoughtfully and addressed all my concerns. I support the publication of the revised manuscript.

Reviewer #3 (Remarks to the Author):

The authors addressed all the comments